# Effect of Heat Stress on Bacterial Composition and Metabolism in the Rumen of Lactating Dairy Cows

**DOI:** 10.3390/ani9110925

**Published:** 2019-11-05

**Authors:** Shengguo Zhao, Li Min, Nan Zheng, Jiaqi Wang

**Affiliations:** State Key Laboratory of Animal Nutrition, Institute of Animal Sciences, Chinese Academy of Agricultural Sciences, Beijing 100193, China; zhaoshengguo1984@163.com (S.Z.); min1988317@163.com (L.M.); zhengnan_1980@126.com (N.Z.)

**Keywords:** rumen, bacterial community, heat stress, dairy cows, metabolism

## Abstract

**Simple Summary:**

Heat stress negatively impacts the health and milk production of dairy cows, and ruminal microbes play an important role in the animal’s milk production. Understanding the link between heat stress and the ruminal microbiome could help to develop strategies to relieve the influence of heat stress by manipulating the ruminal microbial composition. We found that heat-stressed cows had decreased ruminal pH and acetate concentration, whereas the ruminal lactate concentration increased. Heat-stressed cows also had a significantly higher relative abundance of lactate producing bacteria (e.g., *Streptococcus* and unclassified *Enterobacteriaceae*), *Ruminobacter*, *Treponema*, and unclassified *Bacteroidaceae*, all of which utilize soluble carbohydrate as an energy source. The relative abundance of the acetate-producing bacterium *Acetobacter* decreased with heat stress treatment. Therefore, heat stress is associated with changes in ruminal bacterial composition and metabolites, with more lactate and less acetate-producing species in the population, which potentially negatively affects milk production.

**Abstract:**

Heat stress negatively impacts the health and milk production of dairy cows, and ruminal microbial populations play an important role in dairy cattle’s milk production. Currently there are no available studies that investigate heat stress-associated changes in the rumen microbiome of lactating dairy cattle. Improved understanding of the link between heat stress and the ruminal microbiome may be beneficial in developing strategies for relieving the influence of heat stress on ruminants by manipulating ruminal microbial composition. In this study, we investigated the ruminal bacterial composition and metabolites in heat stressed and non-heat stressed dairy cows. Eighteen lactating dairy cows were divided into two treatment groups, one with heat stress and one without heat stress. Dry matter intake was measured and rumen fluid from all cows in both groups was collected. The bacterial 16S rRNA genes in the ruminal fluid were sequenced, and the rumen pH and the lactate and acetate of the bacterial metabolites were quantified. Heat stress was associated with significantly decreased dry matter intake and milk production. Rumen pH and rumen acetate concentrations were significantly decreased in the heat stressed group, while ruminal lactate concentration increased. The influence of heat stress on the microbial bacterial community structure was minor. However, heat stress was associated with an increase in lactate producing bacteria (e.g., *Streptococcus* and unclassified *Enterobacteriaceae*), and with an increase in *Ruminobacter*, *Treponema*, and unclassified *Bacteroidaceae*, all of which utilize soluble carbohydrates as an energy source. The relative abundance of acetate-producing bacterium *Acetobacter* decreased during heat stress. We concluded that heat stress is associated with changes in ruminal bacterial composition and metabolites, with more lactate and less acetate-producing species in the population, which potentially negatively affects milk production.

## 1. Introduction

Heat stress negatively impacts animal production and jeopardizes animal husbandry and the economy [1,2,3]. In particular, dairy cows are extremely susceptible to heat stress [4]. One of the most noticeable production problems associated with heat stress is a pronounced reduction in milk production and quality in lactating dairy cows [5]. Heat stressed dairy cows are likely to become a more widespread problem in the future as the temperatures continue to increase worldwide.

Although the effects of heat stress on dairy cows have been extensively studied, the results and implications cannot be generalized because of the limitations of temperature manipulation in an environmental chamber [6]. Utilizing natural temperature variations and climate anomalies rather than environmental chamber, is much closer to animal production systems. Previously, we demonstrated the influence of seasonal heat stress on dairy cows under natural temperature variations using proteomics and metabolomics [7,8,9,10]. These intriguing studies revealed that plasma lactate and lactate dehydrogenase levels were significantly higher in heat-stressed dairy cows, which might suggest enhanced glycolysis during heat stress [9,10]. The effects of heat stress on bacterial diversity in the rumen of Holstein heifers were previously investigated in a climatic chamber by using a 16S rRNA gene library approach, and the results showed that the diversity of the bacteria was affected by increasing environmental temperature and humidity [11,12]. However, there is relatively little information available about the responses of the ruminal bacterial community in dairy cows to heat stress under natural environmental conditions. The purpose of this study was to profile the changes in the composition and metabolism of bacteria in the rumen of heat-stressed lactating dairy cows.

## 2. Materials and Methods

Experimental procedures involving the care and management of dairy cows were approved by the Animal Care and Use Committee for Livestock of the Institute of Animal Sciences, Chinese Academy of Agricultural Sciences (No. IAS201914).

### 2.1. Animals and Experimental Design

Eighteen lactating Holstein dairy cows were selected at a commercial dairy farm, and divided into two groups. The control group (*n* = 9, 2.1 ± 0.4 parities, 141 ± 16 days in milk (DIM)) was investigated in the cool spring month of June for four days, and the heat stress group (*n* = 9, 2.3 ± 0.5 parities, 144 ± 3 DIM) was investigated during the hot summer month of July for four days. The temperature and humidity were recorded each day at 06:00, 14:00 and 22:00 to calculate the temperature-humidity index (THI). The THI mean values for the control and heat-stress group were 71 (average temperature of 24 °C and humidity of 63%) and 87 (average temperature of 34 °C and humidity of 67%), respectively, which were equal to the cool and heat-stress models [13]. A total mixed ration diet and water were provided ad libitum. The list of dietary ingredients (dry-matter based) was as follows: 23% corn, 16.7% corn silage, 14% alfalfa hay, 9.4% barley, 8.7% Chinese wild rye, 8.3% cottonseed, 7.4% soybean meal, 3.7% dry distillers grains, 3.4% cottonseed meal, 3.4% rapeseed meal, 0.64% sodium bicarbonate, 0.54% dicalcium phosphate, 0.49% salt, and 0.33% vitamin-mineral premix. The diet (dry-matter based) contained 16.9% crude protein, 39.1% neutral detergent fiber, 22.5% acid detergent fiber, 1.07% calcium, and 0.48% phosphorus, with a net energy of lactation of 6.99 MJ/kg. The diet was offered to ensure 5% refusals. All the cows were housed in tie-stalls with access to fresh water during the trial period. Milk production and feed intake were recorded for each cow for four days. Feed intake for individual cows was calculated by subtracting the orts from the feed offered.

### 2.2. Collection of Rumen Fluid Samples

Rumen fluid samples from the control and heat-stressed groups were collected from each cow via esophageal tubing at 14:00 each day for four days. The first hundreds mL of rumen fluid were discarded to minimize saliva contamination [14]. The pH value was immediately measured. Samples were filtered through four layers of cheesecloth, and then stored at −80 °C for the determination of ruminal fermentation parameters and DNA extraction.

### 2.3. Determination of Ruminal Fermentation 

After thawing, ruminal supernatant was obtained by centrifugation (10,000× *g* for 10 min at 4 °C). Volatile fatty acids in supernatant were analyzed by gas chromatography as described by Mohammed et al. [15]. Lactate concentration in supernatant was determined by spectrophotometry using a commercial kit (Nanjing Jiancheng Co., Nanjing, China). 

### 2.4. DNA Extraction and 16S rRNA Gene Sequencing

Microbial DNA was extracted from the rumen fluid samples using a method involving cetyl trimethylammonium bromide and bead beating [16]. DNA concentration was determined using a NanoDrop ND1000 spectrophotometer (NanoDrop Technologies Inc., Wilmington, DE, USA). The V3–V4 region of the bacterial 16S rRNA gene was PCR-amplified (3 min at 95 °C; followed by 30 cycles of 30 s at 95 °C, 30 s at 55 °C, and 45 s at 72 °C; with a final extension of 10 min at 72 °C) using primers 338F (5′-barcode-ACTCCTACGGGAGGCAGCAG-3′) and 806R (5′-GGACTACHVGGGTWTCTAAT-3′) [17]. PCR reactions were performed in triplicate in 20 μL mixtures containing 4 μL of 5× PrimeSTAR buffer, 2 μL of 2.5 mM dNTPs, 0.8 μL of each primer (5 μM), 0.4 μL of PrimeSTAR heat stress DNA polymerase (Takara, Dalian, China), and 20 ng of template DNA. Amplicons were extracted from 2% agarose gels and purified using the AxyPrep DNA gel extraction kit (Axygen Biosciences, Union City, CA, USA), and quantified using the Qubit 2.0 Fluorometric Quantitation. Purified amplicons were pooled in equimolar quantities and paired-end sequenced (2 × 300 bp) using an Illumina MiSeq platform at Majorbio (Shanghai, China).

### 2.5. Sequences Analysis

QIIME (version 1.9) was used for sequencing data analysis. Raw fastq files were de-multiplexed and then quality-filtered using the following criteria: (1) 300 bp reads were truncated at any site with an average quality score <20 over a 50 bp sliding window, discarding truncated reads shorter than 50 bp; (2) exact barcode matching was required, two nucleotide mismatches in primer matching were allowed, and reads containing ambiguous characters were removed; (3) only sequences with overlaps over 10 bp were assembled based on the overlap sequence. Reads that could not be assembled were discarded. 

Operational taxonomic units (OTUs) were clustered with a 97% similarity cut-off using UPARSE (version 7.1; http://drive5.com/uparse/), and chimeric sequences were identified and removed using UCHIME [18]. The OTUs were filtered based on the total observation count of an OTU <10 and the number of samples in an OTU <2 using QIIME [19]. The OTUs were further assigned to taxa using the ribosomal database project classifier based on the Greengenes database 13_8 [20]. Alpha diversity parameters, including Good’s coverage, observed species, Chao l, PD whole tree, Shannon and Simpson indices were calculated for each sample. The weighted UniFrac distance was then calculated and used in principal coordinate analysis in QIIME. Analysis of similarities (ANOSIM) was used to compare the differences in the bacterial community between two groups in QIIME. The linear discriminant analysis effect size (LEfSe) algorithm was used to identify bacteria with different abundance [21]. Sequences generated in the current study have been deposited in the NCBI Sequence Read Archive database (SRA; http://www.ncbi.nlm.nih.gov/Traces/sra/) under the accession number SRP144598.

### 2.6. Statistical Analysis

Differences in pH values, volatile fatty acids and lactate levels were statistically analyzed using the SAS GLM procedure (SAS Institute, Inc., Cary, NC, USA). The data are presented as the least square means ± SEM. Differences were declared significant at *p* < 0.05. Alpha diversity and bacteria abundance difference were evaluated using the Mann Whitney U test for two-group comparisons implemented in GraphPad Prism (GraphPad, Inc., La Jolla, CA, USA). 

## 3. Results

### 3.1. Animal Production and Rumen Fermentation

Compared with the control group, heat stress significantly decreased the dry matter intake and milk production (Figure 1). As shown in Table 1, heat stress resulted in the reduction of pH and acetate levels in the rumen compared with the control group (*p* < 0.01). However, heat stress resulted in an increase in lactate level in the rumen (*p* < 0.05). No significant differences in propionate and butyrate levels were observed between the heat-stressed and control groups.

### 3.2. Ruminal Bacterial Diversity

The results of the alpha diversity index analysis are presented in Figure 2. The Good’s coverage was above 96% in the heat-stressed and control groups, suggesting there was enough data to reflect the changes in the rumen bacterial community. No significant difference between the heat-stressed and control group was apparent in the observed species, PD whole tree, and Shannon and Simpson indices. Chao l was an exception as the heat-stress group showed an increase in the richness index (Chao l) (*p* < 0.05).

Principal coordinate analysis based on the weighted UniFrac distance was performed to compare the bacterial community between the heat-stressed and the control group (Figure 3). Principal coordinates 1 and 2 explained 44.03% and 21.14% of the total variation, respectively. There was no obvious separation between the samples in the two groups. ANOSIM analysis showed that the difference between the heat-stressed and the control group was not significant (*R* = 0.05, *p* = 0.15). 

### 3.3. Ruminal Bacterial Composition

In rumen fluid, we found that *Prevotella* (65.4%) was the predominant genus on average (Figure 4). LEfSe analysis revealed that heat stress resulted in a significant increase in the relative abundances of *Streptococcus* and unclassified *Enterobacteriaceae*, *Ruminobacter*, *Treponema* and unclassified *Bacteroidaceae,* and a reduction in the relative abundances of *Acetobacter* (Figure 5 and Figure 6).

## 4. Discussion

In the current study, heat stress significantly reduced feed intake and milk production in dairy cows, which is in agreement with most previous studies [22]. The reduced feed intake was an important reason for the reduced milk production in heat-stressed dairy cows, however, Wheelock et al. (2010) found that only half of the decrease in milk yield can be accounted for by a decrease in feed intake [23]. This suggests that heat stress could affect milk production by changing the metabolism of nutrients or milk biosynthesis in addition to feed intake in dairy cows. As to the importance of rumen microbes in this process, we hypothesized that heat stress could influence the rumen microbial composition and their metabolism. 

We found that heat stress resulted in reduced pH and increased lactate level in the rumen fluid. Higher production of lactate can decrease energy availability, reduce pH and then inhibit the growth of pH-sensitive ruminal bacteria (e.g., cellulolytic bacteria), and cause sub-acute ruminal acidosis [24]. Sub-acute ruminal acidosis is an important metabolic disturbance in rumen that easily suppresses milk production in dairy cows [25]. In our previous studies, we observed a higher plasma lactate level in heat-stressed cows [9,10]. Therefore, the enriched lactate in rumen could be transferred to the blood, and have a negative effect on animal health [26]. In this study, we also found that heat stress significantly reduced rumen acetate levels. It is well known that lactate and acetate are the main metabolites from soluble carbohydrate and fiber, respectively. Even though we did not estimate the proportion of concentrate intake, Uyeno et al. [12] observed that heat stress changed the feed intake composition significantly, and in particular, increased the intake of more concentrates compared to forage. We inferred that the higher proportion of concentrates in the intake diet could be the reason for changes in the lactate and acetate in rumen of dairy cows under heat stress. In addition, a high proportion of concentrates in the intake could decrease rumen motility and rumination, which lowers saliva production and the ruminal buffering ability under heat stress [27]. Thus, the ruminal pH could be easily changed with more production of lactate.

Rumen metabolites are produced by microbes, therefore the changed metabolites in this study might be due to the altered microbial composition. Here, we found that heat stress had no influence on the alpha and beta diversity of ruminal bacteria, except for the richness. However, we found heat stress led to a significant increase in the abundance of *Streptococcus*, unclassified *Enterobacteriaceae*, *Ruminobacter*, *Treponema* and unclassified *Bacteroidaceae*, all of which utilize soluble carbohydrate as an energy source. *Streptococcus* is the major genus of lactate-producing bacteria in the rumen [28,29], and the majority of bacteria in *Enterobacteriaceae* also produce lactate. Substantial growth in *Streptococcus* and *Enterobacteriaceae* potentially lead to an increase in lactate levels and a reduction of pH in the rumen. In support of this phenomenon, Wang et al. [30] observed that an increase in the abundance of *Streptococcus* in cow rumen is accompanied by an increase in lactate levels and pH reduction. Therefore, increased numbers of lactate-producing bacteria (e.g., *Streptococcus*) resulted in the production of lactate in the rumen under heat stress.

According to Bekele et al. [31] and Liu et al. [32], *Treponema* represented 1.05–3.05% of all rumen bacteria and was one of the core members of the rumen bacterial community. Among these, uncultured *Treponema* species were more abundant than cultured representatives of the *Treponema* genus [33]. *Treponema* is thought to be mainly involved in the degradation of pectin [33]. Bekele, Koike and Kobayashi [31] concluded that *Treponema* participated in the digestion of concentrates. In addition, *Ruminobacter amylophilus,* a representative species from *Ruminobacter,* has a high starch degradation ability in rumen [34]. Therefore, we inferred that the increased abundance of lactate producing or soluble carbohydrate digesting bacteria in heat-stressed dairy cows was potentially due to the increase in the dietary concentrate to roughage ratio for intake. 

In this study, we also found that heat stress led to the reduction of acetate-producing *Acetobacter*. *Acetobacter* has the ability to produce acetate by oxidizing sugars [35]. During feeding, drinking or rumination, a small amount of oxygen is infused into the rumen fluid, and the taking up of oxygen by *Acetobacter* helps to construct an anaerobic environment that allows the anaerobic bacteria and archaea to grow better. Lyons et al. [35] showed a higher relative abundance of *Acetobacter* in the rumen of cows in the mid-lactation period compared with late-lactation. The reduction in *Acetobacter* was consistent with the decrease of acetate in the rumen fluid. It was noticeable that the relative abundance of *Acetobacter* in this study was not high, so its contribution, or that of other bacteria, to the reduction in acetate concentration needs further verification. In addition, changes in other microbes, such as anaerobic fungi, archaea and protozoa, affected by heat stress also needs further study to obtain a comprehensive explanation for rumen microbial fermentation.

It should be noted that the ruminal bacterial composition and metabolism were potentially indirectly affected by heat stress. We know that heat stress can induce changes in several factors, such as decreasing dry matter intake, sorting of preferred rations, decreased cud chewing time and salivary bicarbonate infusion into rumen—these factors are all associated with ruminal bacterial composition and metabolism. However, which factor or factors induced by heat stress are the primary cause of ruminal fermentation changes is still a question that needs to be answered.

## 5. Conclusions

Heat stress affects the ruminal bacterial composition and metabolism, and results in more lactate and less acetate producing bacteria, which may influence the health of cows or milk production. It is necessary to regulate the ruminal microbial fermentation by using nutrients or additives to alleviate the negative effect of heat stress.

## Figures and Tables

**Figure 1 animals-09-00925-f001:**
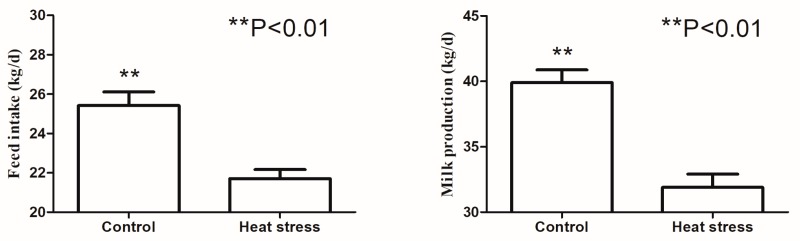
The effects of heat stress on feed intake and milk production in dairy cows.

**Figure 2 animals-09-00925-f002:**
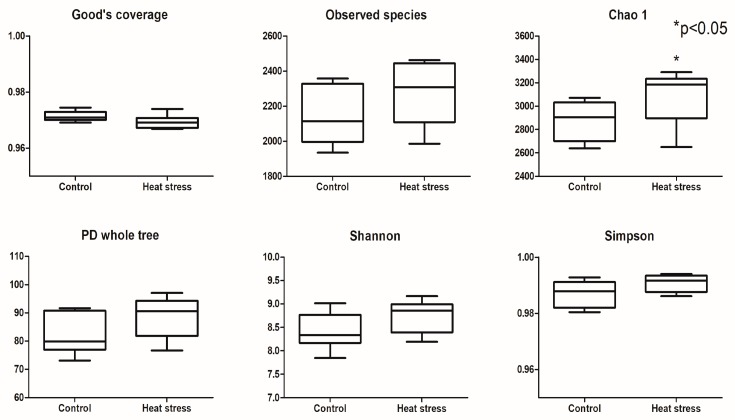
The effects of heat stress on the alpha diversity index of rumen bacteria.

**Figure 3 animals-09-00925-f003:**
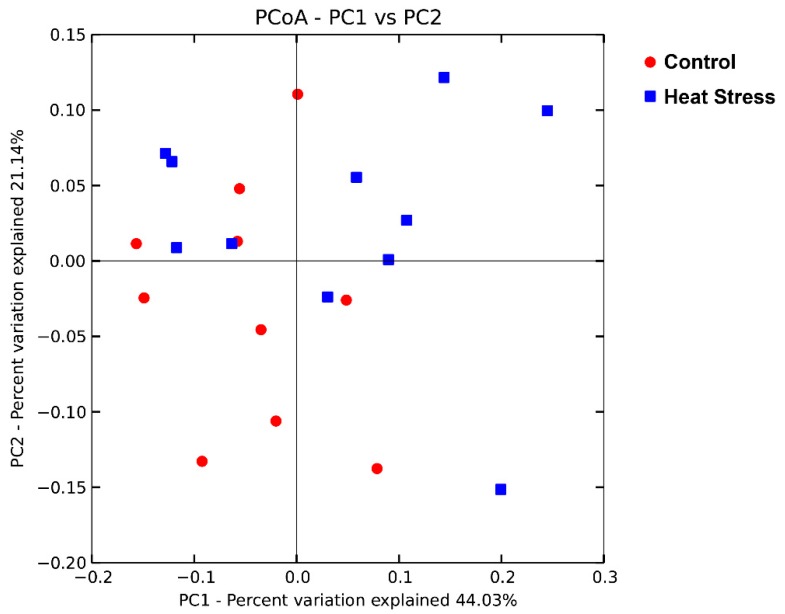
Principal coordinate analysis of the effect of heat stress on the rumen bacterial community based on the weighted UniFrac distance.

**Figure 4 animals-09-00925-f004:**
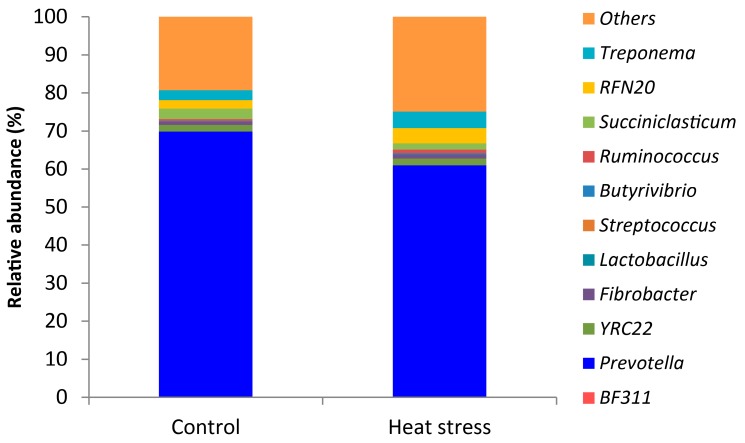
The relative abundant (%) of genera in the rumen of dairy cows with or without heat stress.

**Figure 5 animals-09-00925-f005:**
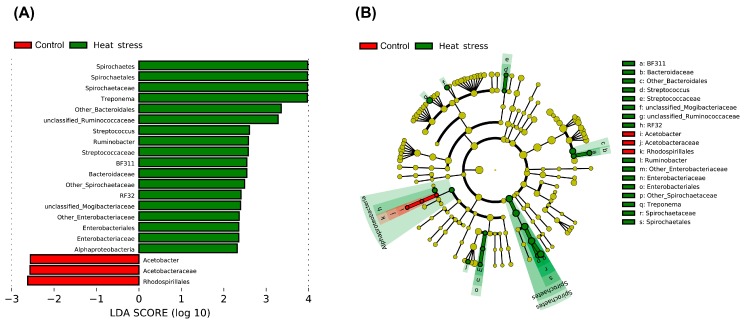
The linear discriminant analysis effect size (LEfSe) of the changes in the rumen bacterial community by heat stress. (**A**) Visualization of differential bacteria with the effect sizes. The length of the bar represents a log10 transformed effect size (LDA) score. The colors represent the group in which that taxa were found to be more abundant compared to the other group. (**B**) Cladogram of differential bacteria. The color represents the branch of the phylogenetic tree that most significantly represents a certain group.

**Figure 6 animals-09-00925-f006:**
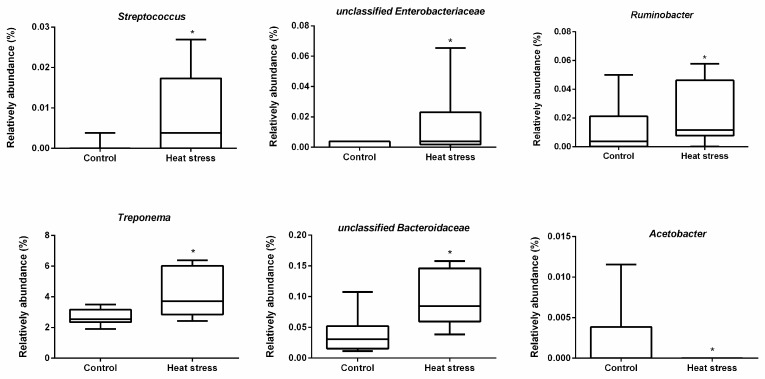
Boxplot of the significant differential rumen bacteria affected by heat stress. * Indicates the significant difference between two groups.

**Table 1 animals-09-00925-t001:** The effects of heat stress on rumen fermentation.

Index	Control	Heat Stress	SEM	*p* Value
pH	6.31	5.89	0.07	<0.01
Lactate (mmol/L)	0.72	2.07	0.31	0.02
Total VFAs (mmol/L)	104.98	94.75	3.02	<0.01
Acetate (mmol/L)	69.55	58.18	2.19	<0.01
Propionate (mmol/L)	24.46	26.53	0.96	0.30
Butyrate (mmol/L)	10.97	10.04	0.40	0.26

## Data Availability

Sequences generated in the current study have been deposited in the NCBI Sequence Read Archive database (SRA; http://www.ncbi.nlm.nih.gov/Traces/sra/) under the accession number SRP144598.

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
