# Peer review of "Effect of Heat Stress on Bacterial Composition and Metabolism in the Rumen of Lactating Dairy Cows"

_animals, 2019, doi:10.3390/ani9110925_

Round 1
Reviewer 1 Report
The authors have adequately addressed reviewers' concerns.
Author Response
Thanks for your comments.
Reviewer 2 Report
The authors have undertaken many of the suggested changes but significant improvement is possible. I have edited the submitted abstract into a form that reflects the necessary changes and if the article is modified to match the edited abstract, it should be accepted for publication.
Suggested edited form of the abstract:
Abstract: Heat stress negatively impacts health and milk production of dairy cows, and ruminal microbial populations play an important role in dairy cattle production. No currently available studies investigate heat stress associated changes in the rumen microbiome of lactating dairy cattle. Improved understanding of linkages between heat stress and the ruminal microbiome may provide benefit in developing strategies for relieving the influence of heat stress on ruminants by manipulating ruminal microbial composition. In this study, we describe the ruminal bacterial composition and metabolites in heat stressed and non-heat stressed dairy cows. Eighteen lactating dairy cows were divided to two groups treatment groups, on with heat stress and one without heat stress. Dry matter intake was measured and rumen fluid from all cows in both groups was collected. The bacterial 16S rRNA genes in ruminal fluid were sequenced, and the rumen pH and the bacterial metabolites lactate and acetate quantified. Heat stress was associated with significantly decreased dry matter intake and milk production. Rumen pH and rumen acetate concentrations were significantly decreased in in the heat stressed group, while ruminal lactate concentration increased. The influence of heat stress on microbial bacterial community structure was minor. However, heat stress was associated with an increase in lactate producing bacteria (e.g. Streptococcus and unclassified Enterobacteriaceae), and with an increase in Ruminobacter, Treponema, and unclassified Bacteroidaceae, all of which utilize the soluble carbohydrates as an energy source. The relative abundance of acetate producing bacterium Acetobacter decreased during heat stress. We concluded that heat stress is associated with changes of ruminal bacterial composition and metabolites, with more lactate and less acetate producing species in the population, which potentially negatively affects the milk production.
Author Response
We have revised according to your suggestion.
Reviewer 3 Report
Dear Authors,
I have only 2 minor comments:
Line 210: we did not detect...…. Sentence is not cmpleted
Line 216: while more production of lactate.......... The sentence is not clear. The acetate production was reduced in your study. However, this sentence shows the other way
Author Response
We have revised according to your suggestion.
This manuscript is a resubmission of an earlier submission. The following is a list of the peer review reports and author responses from that submission.
Round 1
Reviewer 1 Report
Zhao et al. reported the effects of the heat stress on bacterial composition and metabolism in the rumen of dairy cows. This is a relatively well written manuscript, and the manuscript offers some interesting data. However, the conclusions drawn based on the given data analyses are often exaggerated. Explanations for the differences noted in this study are less reasonable without the proper evidence.
A major limitation of this manuscript is the study design. Authors did not present the baseline data which should have been collected before treatments. Authors should show that there were no differences in the microbial compositions and rumen fermentation between groups before treatments. Without these, explanations for the differences noted in this study are less reasonable. This applies to all the data that the authors presented.
For example, the authors can not claim that the heat stress changed pH and acetate levels in the animals without the data that shows no differences of those levels between groups before heat treatment. Contrary to authors’ claim, it is possible that pH and acetate levels in the rumen between groups were already different before heat treatment.
Likewise, the authors do not have evidences to claim that the heat stress induced microbial changes in the rumen. The baseline data which are microbial compositions before treatments should been presented to show that there was no difference in microbial compositions between groups before treatment.
Also, the data presented in this manuscript should be addressed in association with the complex microbial niche which contains a variety of microbes including protozoa and fungi.
Author Response
1. COMMENTS: Zhao et al. reported the effects of the heat stress on bacterial composition and metabolism in the rumen of dairy cows. This is a relatively well written manuscript, and the manuscript offers some interesting data.
RESPONSE: Thanks for your positive comments.
2. COMMENTS: However, the conclusions drawn based on the given data analyses are often exaggerated. Explanations for the differences noted in this study are less reasonable without the proper evidence. A major limitation of this manuscript is the study design. Authors did not present the baseline data which should have been collected before treatments. Authors should show that there were no differences in the microbial compositions and rumen fermentation between groups before treatments. Without these, explanations for the differences noted in this study are less reasonable. This applies to all the data that the authors presented. For example, the authors can not claim that the heat stress changed pH and acetate levels in the animals without the data that shows no differences of those levels between groups before heat treatment. Contrary to authors’ claim, it is possible that pH and acetate levels in the rumen between groups were already different before heat treatment. Likewise, the authors do not have evidences to claim that the heat stress induced microbial changes in the rumen. The baseline data which are microbial compositions before treatments should been presented to show that there was no difference in microbial compositions between groups before treatment.
RESPONSE: We did not collect the samples at the beginning of experiment. As you said, that would be better to do that to minimize the difference of two groups and give a strong conclusion. But we have several solutions to minimize the difference of two groups in our study. Firstly, we just selected the cows in similar parity and days in milk which are the factors influencing rumen or body metabolism largely. Secondly, all cows had the same diet to keep the similar ruminal fermentation at the beginning of trail. Thirdly, the cows were selected and distributed to groups randomly, and the probability of difference for ruminal metabolites and bacterial composition was very small in statistics. So we think it is acceptable to just use the data in the last period of experiment for this study.
Here are some related publications without sampling at the beginning of experiment:
Uyeno Y, Sekiguchi Y, Tajima K, Takenaka A, Kurihara M, Kamagata Y. An rRNA-based analysis for evaluating the effect of heat stress on the rumen microbial composition of Holstein heifers. Anaerobe. 2010;16(1):27-33.
Zhong S, Ding Y, Wang Y, Zhou G, Guo H, Chen Y, Yang Y. Temperature and humidity index (THI)-induced rumen bacterial community changes in goats. Appl Microbiol Biotechnol. 2019;103(7):3193-3203.
3. COMMENTS: The data presented in this manuscript should be addressed in association with the complex microbial niche which contains a variety of microbes including protozoa and fungi.
RESPONSE: As you said, it is better to reveal the complex microbes related to heat stress. We know that the bacteria is the most important microbe due to large number and complex metabolism. So we selected the bacteria as the target for the first time. In our following study, we will detect a variety of microbes including protozoa and fungi that influenced by heat stress. In the revised manuscript, we added the option you said to the discussion. (line 247-249)
Reviewer 2 Report
This article represents a great deal of interesting work in a field of emerging interest. The quantification of rumen microbiota related to heat stress is useful in understanding decreased production related to heat stress, but the authors assertion the heat stress induced the changes is not supported. Rather, the changes are associated with heat stress, but more likely caused by other related factors, including decreased dry matter intake, potential preferential sorting of the ration by heat stressed individuals, decreased cud chewing times resulting from decreased feed intake, and likely decreased salivary bicarbonate presented to the rumen.
The paper lacks a strong control group, which is understandable given the goal of evaluation in a production unit. This factor should be emphasized in the introduction and discussion.
Inclusion of the CHAO 1 analysis adds little to the paper, and I question the necessity of presenting it.
English language corrections, particularly including choice of the appropriate verb tense, is strongly recommended prior to publication.
Author Response
1. COMMENTS: This article represents a great deal of interesting work in a field of emerging interest. The quantification of rumen microbiota related to heat stress is useful in understanding decreased production related to heat stress.
RESPONSE: Thanks for your positive comments. As we know, this is the first report to study the effect of heat stress on ruminal bacterial composition and metabolism for lactating dairy cows under natural environment.
2. COMMENTS: The authors assertion the heat stress induced the changes is not supported. Rather, the changes are associated with heat stress, but more likely caused by other related factors, including decreased dry matter intake, potential preferential sorting of the ration by heat stressed individuals, decreased cud chewing times resulting from decreased feed intake, and likely decreased salivary bicarbonate presented to the rumen. The paper lacks a strong control group, which is understandable given the goal of evaluation in a production unit. This factor should be emphasized in the introduction and discussion.
RESPONSE: We are agreed to your option. We found the ruminal bacterial composition and metabolism could be influenced by heat stress. Actually heat stress is not the direct factor for the changes of ruminal fermentation. Normally heat stress influences ruminal fermentation by affecting feed intake, cud chewing, saliva infusion to rumen, etc. In our revised manuscript, we changed the description in title, abstract and discussion to avoid understanding of direct influence for heat stress. We also add your option to our discussion section. (line 250-255)
Here are some related publications without feed intake or saliva control:
Uyeno Y, Sekiguchi Y, Tajima K, Takenaka A, Kurihara M, Kamagata Y. An rRNA-based analysis for evaluating the effect of heat stress on the rumen microbial composition of Holstein heifers. Anaerobe. 2010;16(1):27-33.
Zhong S, Ding Y, Wang Y, Zhou G, Guo H, Chen Y, Yang Y. Temperature and humidity index (THI)-induced rumen bacterial community changes in goats. Appl Microbiol Biotechnol. 2019;103(7):3193-3203.
3. COMMENT: Inclusion of the CHAO 1 analysis adds little to the paper, and I question the necessity of presenting it.
RESPONSE: Chao1 is not an important parameter but we still want to keep it. This can tell people how changes of bacterial richness.
4. COMMENT: English language corrections, particularly including choice of the appropriate verb tense, is strongly recommended prior to publication.
RESPONSE: We have revised the manuscript language according your suggestion.
Reviewer 3 Report
Dear authors,
In this study, authors studied “Heat stress induced changes of bacterial composition and metabolism in the rumen of dairy cows”. Although there have been several studies investigated the bacterial community changes during heat stress, the current study was well written. I have only some minor suggestions in this study.
Material and method: More information about the environmental temperature and relative humidity during these two periods. Although THI was given authors will be interested to know temperature differences between these two periods. Material and method: more information of the management condition; individual or loose housing, how the feed intake measurement was taken? Discussion: some discussion about the relationship among heat, feed intake, rumination activity and pH drop.Some specific comments
Line 226- digestion of concentrates Line 237: The relative abundance of Acetobacter in the rumen is low. Do you think this group will be responsible for the reduction of acetate concentration?Author Response
1. COMMENT: In this study, authors studied “Heat stress induced changes of bacterial composition and metabolism in the rumen of dairy cows”. Although there have been several studies investigated the bacterial community changes during heat stress, the current study was well written. I have only some minor suggestions in this study.
RESPONSE: Thanks for your positive comments.
2. COMMENT: Material and method: More information about the environmental temperature and relative humidity during these two periods. Although THI was given authors will be interested to know temperature differences between these two periods. Material and method: more information of the management condition; individual or loose housing, how the feed intake measurement was taken?
RESPONSE: We have added these description to the revised manuscript. (line 74-75, 82-85)
3. COMMENT: Discussion: some discussion about the relationship among heat, feed intake, rumination activity and pH drop.
RESPONSE: We have added these description to the revised manuscript. (line 212-216)
4. COMMENT: Line 226- digestion of concentrates
RESPONSE: We have revised it according to your suggestion. (line 234)
5. COMMENT: Line 237: The relative abundance of Acetobacter in the rumen is low. Do you think this group will be responsible for the reduction of acetate concentration?
RESPONSE: The relative abundance of Acetobacter is low, but we do not know the activity of acetate producing of this bacterium. So it is hard to give conclusion for its contribution for the reduction of acetate concentration. We added these to the revised manuscript. (line 246-247)